# Localization and Classification of Venusian Volcanoes Using Image Detection Algorithms

**DOI:** 10.3390/s23031224

**Published:** 2023-01-20

**Authors:** Daniel Đuranović, Sandi Baressi Šegota, Ivan Lorencin, Zlatan Car

**Affiliations:** 1Rijeka Development Agency PORIN, Ul. Milutina Barača 62, 51000 Rijeka, Croatia; 2Faculty of Engineering, University of Rijeka, Vukovarska 58, 51000 Rijeka, Croatia

**Keywords:** artificial intelligence, convolutional neural network, object detection, YOLO, venusian volcanoes, Magellan data set

## Abstract

Imaging is one of the main tools of modern astronomy—many images are collected each day, and they must be processed. Processing such a large amount of images can be complex, time-consuming, and may require advanced tools. One of the techniques that may be employed is artificial intelligence (AI)-based image detection and classification. In this paper, the research is focused on developing such a system for the problem of the Magellan dataset, which contains 134 satellite images of Venus’s surface with individual volcanoes marked with circular labels. Volcanoes are classified into four classes depending on their features. In this paper, the authors apply the You-Only-Look-Once (YOLO) algorithm, which is based on a convolutional neural network (CNN). To apply this technique, the original labels are first converted into a suitable YOLO format. Then, due to the relatively small number of images in the dataset, deterministic augmentation techniques are applied. Hyperparameters of the YOLO network are tuned to achieve the best results, which are evaluated as mean average precision (mAP@0.5) for localization accuracy and F1 score for classification accuracy. The experimental results using cross-vallidation indicate that the proposed method achieved 0.835 mAP@0.5 and 0.826 F1 scores, respectively.

## 1. Introduction

The development of the internet and hardware resources in the last twenty years has facilitated access to a large amount of data. The data generated in recent years is expanding in variety and complexity, and the speed of collection is also increasing. Such exponential growth in the amount of data requires simple, fast, and accurate tools for processing. Artificial intelligence (AI) jumps in as a solution. Computer vision is one of the most attractive research areas today. Scientists in this field have achieved enormous progress in recent years, from the development of an algorithm for driving autonomous vehicles [1], all the way to the detection of cancer using only information from images [2].

In the past, data collection in astronomy was a manual and time-consuming job for humans, but with the advancement of technology and automation, today the amount of data collected by observing the sky is almost unimaginable. Looking back 20 years, efforts such as the Sloan Digital Sky Survey (SDSS) [3], Pan-STARRS [4], and the Large Synatopic Survey Telescope (LSST) [5], have shifted from individualized data collection to studying larger parts of the sky and wider wavelength ranges of light, e.g., collecting data on more events, objects, and larger areas of the sky through larger areas and better light detectors. SDSS is one of the largest astronomical surveys today. Every night the SDSS telescope collects 200 GB of data, while there are telescopes that collect up to 90 terabytes of data every night (Table 1) [6]. Due to the enormous amount of data collected, astronomers struggle with more than 100 to 200 petabytes of data per year, which requires a large storage infrastructure. Therefore, it is crucial to process this large amount of collected data. Such an exponential increase in data provides an ideal opportunity to apply artificial intelligence and machine learning to help process all this data.

The application discussed in this paper focuses on data collected from the Magellan spacecraft, which was launched in 1989 and arrived in the orbit of Venus in 1990. During its four years in orbit around Earth’s sister planet, Magellan recorded 98% of the surface of Venus and collected the gravitational data of the planet itself in high resolution using synthetic aperture radar (SAR) [7]. During Magellan’s active operation, about 30,000 images of the surface of Venus were collected, on which there are also numerous volcanoes, but the manual labeling of these images is time-consuming and so far, only 134 images have been processed by geological experts [8]. It is believed that there are more than 1600 large volcanoes on the surface of Venus whose features are known so far, but their number may be over a hundred thousand or even over a million [8,9]. Understanding the global distribution of the volcanoes is critical to understanding the geologic evolution of the planet. Therefore, collecting relevant information, such as volcanoes size, location, and other relevant information, can provide the data that could answer questions about the geophysics of the planet Venus. The sheer volume of collected planetary data makes automated tools necessary if all of the collected data are to be analyzed. It was precisely for this reason that a tool was created using machine learning algorithms that would serve as an aid in labeling such a large amount of data.

Burl et al. proposed the JARtool System (JPL Adaptive Recognition Tool) [8], which is a trainable visual recognition system developed for this specific problem. JARtool uses a matched filter derived from training examples, then via principal components analysis (PCA) provides a set of domain-specific features. On top of that, supervised machine learning techniques are applied to classify the data into two classes (volcano or non-volcano). For detecting possible volcanic locations, JARtool’s first component is the focus of attention (FOA) algorithm [10], which outputs a discrete list of possible volcanic locations. Later mentioned, the investigation is concluded using a combination of classifiers in which each classifier is trained to detect a different volcano subclass. In their experiments, each label is counted as a volcano regardless of the assigned confidence (volcanoes are classified depending on their confidence score). Experiments were performed on two types of Magellan data products, Homogeneous and Heterogeneous images. Homogeneous images represent images that are relatively similar in appearance (homogeneous) and are mostly from the same area. Heteregoeneus images represent images chosen from random locations, and those images contained a significantly greater variety in appearance. The JARtool system performed relatively well on homogeneous image sets in the 50–64% range of detected volcanoes (one class, regardless of the classification), while performance on heterogeneous image sets was in the 35–40% range.

The scope of this problem is focused on detecting geological features over the SAR imagery, which can be noisy and thus challenging. Due to the nature of SAR images, they are affected by multiplicative noise, which is also known as speckle noise [11]. This kind of noise is one of the main things that affect the overall performance of any classification methodology. Many studies have applied SAR imagery to machine learning-based algorithms, even including deep learning [12]. The first attempts using deep learning methods in SAR imagery were for a variety of tasks, which includes terrain face classification [13], object detection [14], and even disaster detection [15]. This problem requires both detection and classification of the desired object, which requires the usage of more known convolutional neural network (CNN) architectures such as Fast-RCNN [16], You only look Once algorithm, etcetera. After some research, it is concluded that most of the object detection in SAR imagery is focused on detecting human-made objects. For example, there are deep learning methods used for detecting ships, airplanes, and other man-made features in SAR imagery.

In this paper, authors used the You Only Look Once (YOLO) version 5 algorithm [17] as a base for an automatic tool for localizing and classifying Venusian volcanoes. YOLOv5 algorithm is currently one of the fastest and most precise algorithms for object detection and, today, it is used in almost every field. From the many studies concluded in the past years using YOLO on different problems, it demonstrates better results in more cases compared to other object detectors. For example, Mostafa et al. [18] trained YOLOv5, YOLOX, and Faster R-CNN models for the detection of autonomous vehicles. YOLOv5, YOLOX, and Faster R-CNN achieved mean average precision (mAP) at 0.5 thresholds of 0.777, 0.849, and 0.688, respectively; therefore, YOLO outperforms Faster R-CNN. Wang et al. [19] used the YOLOv5 algorithm in shape wake detection in SAR imagery and achieved an mAP score in the range of 80.1–84.8%, proving that YOLOv5 can successfully be used in complex conditions, such as different sea states and multiple targets. Xu et al. [20] trained the YOLOv5 algorithm for ship detection in large-scene Sentinel-1 SAR images, effectivly scoring mAP of 0.7315. Yoshida and Ouchi also used YOLOv5 for ship detection cruising in the azimuth direction and achieved an mAP score of 0.85 [21]. Reviewing the usage of YOLOv5 in SAR imagery proves that YOLOv5 can successfully be used to detect desired features in such complex imagery as SAR imagery. Furthermore, the mAP score in the range 0.7–0.85 seems to be a good score for detection in SAR imagery.

The goal of this paper is to examine a YOLOv5 algorithm as a backbone to an automation system for labeling small Venusian volcanoes over SAR imagery. Furthermore, this paper focuses on Magellan heterogeneous image sets, because it appears to be a larger problem when reviewing Burl et al. [8] and trying to classify all four classes of volcanoes instead of just treating them as the same class. One of the problems with the Magellan dataset is that it contains a relatively small number of labeled images. Today deep learning models require a large number of images for successful training. Based on the idea of this investigation and the literature overview, the following questions were derived:Is it possible to achieve good classification accuracy with the YOLOv5 algorithm on SAR imagery and such a small dataset?Can classical training data augmentation techniques improve classification accuracy, or are some advanced methods are required?

The structure of this paper can be divided into the following sections: Materials and Methods, Results and Discussion, and Conclusions. In the Materials and Methods section, the dataset description is given as well as the used augmentation methods for training the YOLOv5 algorithm. In Results and Discussions, the results of conducted experiments are presented and discussed. The Conclusion section includes thoughts and conclusions that were obtained during the examination of the results of the experiment and provides answers to the hypotheses defined in the Introduction section.

## 2. Materials and Methods

In this section, the dataset and its preparation for the YOLOv5 algorithm are explained, as well as the utilized augmentation techniques. The steps taken to properly train a YOLOv5 algorithm on a Magellan dataset are next:Formatting the dataset into appropriate YOLO format from a Magellan format;Splitting the dataset into training and testing sets;Performing the augmentation on a training set to increase the number of training images.

All of the above mentioned steps are more described in the following text.

### 2.1. Dataset Description

The Magellan data set consists of 134 images with dimensions of 1024 × 1024 pixels representing the surface of planet Venus. Figure 1 shows a 30 km × 30 km surface image of Venus taken by the Magellan spacecraft. Volcanoes on the surface of the planet can be recognized by their so-called “radar signature”, which is a product of the SAR radar, e.g., the side of the volcano facing the radar is brighter, while the opposite side of the volcano is darker.

The reason for this is that the closer side of the object (in this case the volcano), which is facing the direction of the radar, reflects a larger amount of energy back toward the radar sensor, while the farther (darker) side facing the opposite direction scatters the beam into the surrounding space. The brightness of each pixel is proportional to the logarithm of the reflected energy back to the sensor. Accordingly, the typical “radar signature” of a volcano appears as a “light-dark” circular outline with bright pixels in the center [8]; this described phenomenon is illustrated in Figure 2. Brighter pixels in the very center of the volcano usually appear because the SAR beam is scattered within the crater itself. However, if the crater is small compared to the resolution of the image, it is possible that this feature will not appear. The phenomena described above are just some of the features according to which experts have classified volcanoes on the surface.

Due to the forested nature of SAR images, even experts cannot determine with 100% certainty whether one of the objects in the image is actually a volcano or not. Unfortunately, this problem occurs due to various influencing factors such as image resolution or signal-to-noise ratio (SNR). Different experts will assign different classes to the same volcanoes for the same image. Based on extensive discussions with experts, Burl et al. [8] propose the next five categories of volcanoes:“Category 1—almost certainly a volcano p≈0.98); the image clearly shows a summit pit, a bright-dark pair, and a circular planimetric outline.Category 2—probably a volcano (p≈0.8); the image shows only two of the three category 1 characteristics.Category 3—possibly a volcano (p≈0.6); the image shows evidence of bright-dark flanks or a circular outline; the summit pit may or may not be visible.Category 4—a pit (p≈0.5); the image shows a visible pit but does not provide conclusive evidence for flanks or a circular outline.Category 5—not a volcano (p≈0)” [8].

Where the letter p indicates the probability of a volcano. Figure 3 shows examples of each category of volcanoes proposed by Burl et al. [8].

### 2.2. Dataset Preparation

All images from the data set come in two files of different formats (SDT and SPR) that describe the image. The SPR file is a so-called header file that contains features that describe the image (e.g., image dimensions), while the SDT file contains the binary record of the image itself. Because of this, there is a need to transfer images to a format suitable for learning the YOLO algorithm (e.g., JPEG or PNG format). On this occasion, the vread.py script was developed. Its purpose is to load images from the given two files into the Python environment and return them in PNG format, which is suitable for learning the algorithm. Each image from the data set also contains the so-called ground-truth file of the same name, which indicates the location of the volcano in the image and its belonging to a certain class. Ground-truth files are also called labels (in TXT format). In the Magellan data set, the ground truth comes in the LXYR format, the structure of which is shown in Table 2. The first digit indicates the class of the volcano, the second digit indicates the *x*-coordinate of the origin (in pixels) of the circle, the third indicates the *y*-coordinate of the origin of the circle, and the last digit indicates the radius of the circle.

When marking images, the coordinate system is located in the upper left corner of the image itself, an example of the image coordinate system is shown in Figure 4.

An example of a label for the YOLO format is shown in Table 3 where the first digit indicates the class of the volcano, the second digit indicates the *x*-coordinate of the origin, the third indicates the *y*-coordinate of the origin, and the last two digits indicate the width and height of the bounding box that describes the object normalized to the values [0, 1].

Examples of the difference between the above two formats are shown in Figure 5, where the basic difference between the two object annotation formats can be observed. The Magellan annotation format expresses its center coordinates in pixels, while in the YOLO format, they are normalized. Another visible difference is that the Magellan format uses a circle to mark the object in the image, while the YOLO format uses a rectangle or square to mark it. The last but important difference in the annotations is that, in the YOLO format, class belonging marks start from the number zero, while in the Magellan format, they start from the number one.

Due to the sheer difference in annotations, there was a need to develop a way to convert the labels from the Magellan format to the YOLO format, which is suitable for learning. For this purpose, a script was created. This function loads LXYR format files and converts them to YOLO format as shown in Figure 6.

As already mentioned in the introduction of this chapter, the Magellan data set consists of a total of 134 images with dimensions of 1024 × 1024 pixels. The data set consists of a relatively small number of images, which presents separate challenges when successfully training the model. In Figure 7, a graphic representation of the data set analysis is shown separately for each class. The first column shows the number of instances per class, i.e., the number of objects per class in the data set. The second column shows the average dimensions of an object of that class, while the third column shows how much average image area these objects occupy. From the graph, it can be concluded that the data set is rather unbalanced, i.e., certain object classes are overrepresented, while some classes are underrepresented. There are a total of 1520 marked volcanoes in the Magellan data set, of which 574 (37.76%) are from class four and 510 (33.55%) from class three, while from classes one and two there are 143 (9.41%) and 293 (19.28%) volcanoes. In the second case, it can be observed that objects from class one are the largest, while objects from class four are very small (≈20 × 20 pixels) and occupy an average of 2% of the total image area, which presents some difficulties in the detection and classification of such objects.

### 2.3. Dataset Augmentation

Due to the relatively small number of images in the Magellan data set, a sufficient number of images could not be distributed in the training set to effectively train the model. The number of images in the training set can be artificially enlarged by augmentation techniques. Augmentation is the process of artificially increasing the amount of new data from existing data [22,23]. This includes adding minor changes to the data or using machine learning models to generate new data or geometrical transformations. A schematic representation of data augmentation is shown in Figure 8.

When augmenting the data set, care must be taken to avoid the so-called data leak, i.e., the order between the division of the data set and the augmentation is important. If data augmentation is done before dividing the data set, there is a problem where two versions of the same image (containing the same features) are found in the training and testing sets. If this happens, the results of such a model are not valid and should be approached with caution. During the augmentation of the training set, an attempt was made to make meaningful augmentations, i.e., manipulation of the images was done in such a way that the new artificially obtained images sufficiently represent the environment in which volcano detection is done. During image processing, some examples of augmentation techniques are rotation, mirroring, contrast change, zooming, translation, etc. [22]. The authors used geometric transformations in the augmentation of satellite images, for the reason that objects in satellite images are viewed from above, so transformations such as mirroring across the horizontal or vertical axis, and rotations generally do not add unwanted features to the training set. When augmenting the training data set, the following methods were used:Rotations for 90°, 180°, and 270°;Mirroring across horizontal and vertical axes;Change of contrast, brightness;Cutout technique.

In Figure 9, some of the augmentation techniques used are shown (points 1–3).

If some of the images from the Magellan data set are looked at, it can be noticed that some images have “black zones”. They arise when the radar failed to catch the return rays or when part of the planet’s surface was not in range. Figure 10 shows an example image with a “black zone”.

To simulate this phenomenon, a cutout() function was implemented which draws black rectangles of random dimensions and in random locations. Then the same function checks according to the label files (annotations) whether it covered more than 50% of the surface of a volcano. If the black rectangle covers more than 50% of the surface of a volcano, it is deleted from the list of annotations. An example of this augmentation technique is shown in Figure 11.

#### Augmentation Pipeline

Since it was not possible to artificially increase the number of images to a sufficient number with classic augmentation methods, a more advanced augmentation technique had to be used. To obtain a sufficient number of images and objects for model training, in addition to the classic augmentation technique, mosaic augmentation was used. In Figure 12, a schematic representation of the implementation of the augmentation pipeline using mosaic augmentation is presented. In addition to the classical augmentation of the data set, mosaic augmentation was serially added to artificially increase the number of images. The reason for using classical augmentation in this series is to maximize the number of images before entering mosaic augmentation to obtain as diverse several images as possible.

The mosaic augmentation technique works on the principle of artificially generating a new image from two or more images (usually 4), which is composed of smaller parts of the previous images. This augmentation method is not always convenient to use because it can generate new images that have some new features that are not representative of the cases of the environment in which the model will operate. In this case, this augmentation technique is suitable for use. An example of mosaic augmentation is shown in Figure 13.

### 2.4. YOLOv5 Algorithm

YOLOv5 is a family of one-stage architectures for real-time object detection developed by Ultralytics [17]. This version of the algorithm is based on the original YOLO algorithm developed by Redmon et al. [24], and is currently one of the most well-known and used architectures. Before the creation of YOLO, CNNs with two-stage architectures were used in computer vision. Two-stage networks divide an input image into bounding boxes, then run a classifier on the bounding boxes, and remove the duplicate detections. YOLO algorithm works on the principle of dividing the input image into a matrix with S × S cells of equal dimensions Figure 14. Each of these S cells is responsible for the detection and localization of the object it contains. Accordingly, in addition to the localization of the object, these cells also perform the classification of the object within them, and therefore the position of the object within the cell and its predicted class with probability are obtained as an output. Such a working principle is a one-stage architecture and results in less computing time.

#### YOLOv5 Structure

The architecture of the YOLOv5 algorithm consists of three main components: backbone, head, and neck. The architecture of the YOLOv5 model is shown in Figure 15.

The backbone layer is mainly used to extract important features from a given input image. The backbone of the YOLOv5 architecture is called CSPDarknet-53, and it contains 23 residual units [25]. Residual units are made from one CNN layer with one 3 × 3 layer and one 1 × 1 layer. After the convolutional layer comes a batch normalization layer, followed by an activation function. The YOLOv5 CSPDarknet-53 backbone is pretrained using the ImageNet database [26]. After the backbone layer, comes the neck with path aggregation network (PANet) [27]. PANet adopts a new feature pyramid network structure with an improved bottom-up path that improves the diffusion of low-level features [28]. PANet improves the localization signals in lower layers, which results in a better prediction of the object’s position. The last part of the YOLOv5 structure is the head. The head uses a feature pyramid network (FPN) to detect objects at three different scales [25]. In short, the head layer generates three feature maps of different sizes that are used for multi-scale prediction effectively allowing a model to find large to small-sized objects on images.

### 2.5. Evaluation Method

The detected object’s result can be divided into three possible outcomes. The first outcome defines the bounding box with correct detection, which means that the detected object is identified as true positive (*TP*). The second outcome defines a bounding box with incorrect detection, which is considered a false positive (*FP*), and the third possible outcome defines that the object is not detected with the bounding box (false negative (*FN*)). Based on the three possible outcomes, there are two evaluation metrics defined, precision (*P*) and recall (*R*) [29]. Precision and recall can be calculated using the following equations:(1)$$P=\frac{TP}{TP+FP}$$
(2)$$R=\frac{TP}{TP+FN}$$

A model is trained well if it has high Precision and Recall metrics. An ideal model has zero *FN* predictions and zero *FP* predictions, which means that both Precision and Recall are equal to one.

Using the above-mentioned parameters, the F-measure can be calculated. F-measure (commonly known as *F*1-score) [30] is the localization detection performance, which can be expressed using the following equation:(3)$$F1=2\frac{P\cdot R}{P+R}$$

The next measure that is used is intersection over union (*IoU*) [31]. The *IoU* evaluates the degree of overlap between the predicted effectively box and the bounding box is considered to be an accurate prediction, and is defined by the following expression:(4)$$IoU=\frac{A\cap B}{A\cup B}$$
where:*A* represents the predicted bounding box;*B* represents the correct prediction.

*IoU* takes values between zero and one, where zero represents no overlap, and one represents the complete overlap between two bounding box frames. This measure is useful by setting a threshold value (for example, a threshold value α) and using that threshold it can be decided whether the detection made by the model is correct or not.

Average precision, or AP@α, is the area under the Precision–Recall (*PR*) curve evaluated at the threshold value α. Mathematically, it is defined by the following expression:(5)$$\mathrm{AP}@\alpha =(\int_{0}^{1}p\left(r\right)\;dr)$$
where:AP@α represents average precision;*p*(*r*) represents the *P–R* curve function;α represents a treshold value.

AP@α means that the average precision is evaluated at a threshold value (*IoU* threshold) equal to α. If there are measurements written as AP50 or AP75, they only mean that AP is calculated for values of *IoU* = 0.5 or *IoU* = 0.75. AP is calculated individually for each class. This means that there are as many AP values as there are classes. To obtain an estimate of the precision of the model overall for all classes, the mean value of APs of all classes is calculated and this metric is called the mean average precision (*mAP*) [32]. *mAP* is defined by the following expression and is one of the main metrics used to evaluate model performance:(6)$$mAP@\alpha =\frac{1}{n}\sum_{n=1}^{n}AP_{i}$$
where:mAP@α represents mean average precision;*n* represents number of classes;*AP*_i_ represents average precision for a given class i.

### 2.6. K-Fold Cross Validation

Because of the relatively small dataset, standard practices such as splitting the dataset into two parts for training and testing is not sufficient for providing relative results for model estimation. To provide truthful model estimation, cross-validation was used.

The k-fold cross-validation (KCV) technique is one of the most used approaches for model estimation. The KCV consists of splitting a dataset into k subsets, then iteratively some of them are used to train the model, while the others are used to assess the model’s performance [33,34,35]. In practice, the typical choice of k is between 5 and 10. Yung [36] mentions that k < 5 might cause an issue and thus not give reliable model estimation. In this paper, because of the above mentioned reasons, the 5-fold cross-validation is used. The initial dataset is randomly divided into five equal folds (k = 5), where four folds of the dataset (k − 1) will be used for the training of the model, and one fold will be used for testing the model. This procedure is repeated five times, where each time a different fold is used for model testing (Figure 16).

After finishing experiments, models are estimated using average values and standard deviation from all experiments for mAP, Precision, Recall, and *F*1 score values. For example, the average value for Precision is calculated using:(7)$$P=\frac{1}{N}\sum_{i=1}^{N}P_{i}$$
where:*P* represents average precision from all experiments;*N* represents number of experiments;*P_i_* represents precision for a given experiment *i*.

Standard deviaton for Precision metric can be calculated using:(8)$$\delta \left(P\right)=\sqrt{\frac{1}{N-1}\sum_{i=1}^{N}{\left(P_{i}-P\right)}^{2}}$$
where:*δ* (*P*) represents standard deviation of precision from all experiments;*N* represents number of experiments;*P_i_* represents precision for given experiment *i*;*P* represents the average value of Precision from all experiments.

Average values and standard deviation from all other metrics are calculated using the same principle.

## 3. Results

In this section, the results of the experiments using the YOLOv5 algorithm with cross-validation on the Magellan dataset are presented. Classification accuracy on a validation set is presented using previously described metrics such as mAP, *F1* score, Precision, and Recall. Furthermore, Appendix A, Appendix B and Appendix C contains Precision–Recall (P-R) curves from all trained models using the cross-validation method.

Before starting training and evaluating the results of the model, it is necessary to determine the basic model based on which, by changing hyperparameters and data augmentation techniques, comparison and assessment of the accuracy of the investigated models will be made. The pretrained YOLOV5l6 model was used as the base model with the transfer learning method. YOLOv5l6 weights were pretrained on a COCO dataset [37]. The reason for using transfer learning is that with randomly generated weight coefficients, due to the relatively small number of images, it would not be possible to obtain a sufficiently precise model, and this method has proven to be a good choice for relatively small datasets in the past. Hyperparameters of the YOLOv5l6 model are given in Table 4.

First, the base model with pretrained weights, and without performing any augmentation techniques on a training set was trained for reference. The results of cross-validation metrics for the base model can be seen in Table 5 and Appendix A.

After calculating the average value and standard deviation from all cross-validation experiments performed on a base model, the evaluation of a base model can be seen in Table 6. Furthermore, Figure A1, Figure A2, Figure A3, Figure A4 and Figure A5 contains *P*–*R* curves of the trained base model.

Next, the same cross-validation procedure was performed on a base model, but this time using classical augmentation techniques on a training set. Results of cross-validation metrics for the model can be seen in Table 7.

After calculating the average value and standard deviation from all cross-validation experiments performed on a base model with classical augmentation techniques, the evaluation of a model can be seen in Table 8. Furthermore, Figure A6, Figure A7, Figure A8, Figure A9 and Figure A10 contains *P*–*R* curves of the model trained using classical augmentation techniques.

Finally, for the last cross-validation procedure, the proposed augmentation pipeline technique was used on a training set. The results of cross-validation metrics for the model can be seen in Table 9.

After calculating the average value and standard deviation from all cross-validation experiments performed on a base model with classical augmentation techniques, the evaluation of a model can be seen in Table 10. Moreover, Figure A11, Figure A12, Figure A13, Figure A14 and Figure A15 contain *P*–*R* curves of the model trained using the proposed augmentation pipeline.

## 4. Discussion

From Table 6 it can be seen that the base model after performing cross-validation scores the mean average accuracy value of mAP = 28.4%, which is low accuracy due to the extremely small number of images for training the model. This is why augmentation techniques were used on the training data set. Due to the relatively poor accuracy of the base model, classic augmentation methods were used and tested.

By using classical methods of augmentation over images in the training set and performing cross-validation, Table 8 shows that there is an increase in the accuracy of the model by 2.5% versus the base model. Unfortunately, there is no drastic increase in precision and the obtained results are still unsatisfactory. The reason for this is still the relatively small number of images for training a model with such a large number of parameters. Although the number of images in the training set has been artificially increased by a factor of 8, it is still an insufficient number to successfully train the model. More than 1500 images per object and more than 10,000 total labeled objects per class are needed for successful model training [38]. Since in this case most of the images contain all the objects, for successful training it is enough to artificially increase the number of images to approximate that number. Unfortunately, using classic augmentation techniques, the number of images can be artificially increased only up to a certain limit (it can be increased by a factor of how many techniques are used). Another problem stems from the fact that the Magellan data set has an extremely small number of images (134 in total), and depending on the data set division, the model can be trained well. When dividing the data set, care had to be taken to take enough images for validation and testing to cover a large number of real-world cases, while also allowing a large enough number of images to train the model. For this reason, a more advanced data augmentation technique was used, such as implementing of the above-described augmentation pipeline.

By comparing the results from Table 6, Table 8, and Table 10, it can be observed that the accuracy of the proposed model compared to the base model trained by the proposed increased by 55.1% versus the base model, and 52.6% versus the model trained with classical augmentation techniques, which is proof that this is a successful method for model training. Despite all that, the model gave satisfactory results for the detection and classification of volcanoes over SAR imagery on the surface of the planet Venus.

### Testing the Developed Model

After training the YOLOv5l6 model using the methods described in the previous subsections, the weights that give the best results on testing were downloaded and tested. The detection results of the YOLOv5l6 model are shown in Figure 17, Figure 18 and Figure 19. From Figure 17, it can be noticed how the correct detection of the classification of most objects occurs correctly.

From Figure 18 it can be noticed that the YOLO algorithm has a problem with the detection and classification of objects in places where there are many instances of various classes.

Figure 19 shows one of the cases where class K3 volcanoes were wrongly classified (as class K2) due to their similar characteristics between classes.

From Figure 17, Figure 18 and Figure 19 it can be noticed that the model has successfully detected and classified the majority of volcanoes. Despite everything, the model gave satisfactory results of volcano detection and classification from satellite SAR imagery of the surface of Venus, and it can be concluded that a volcano detection system has been successfully developed and tested.

## 5. Conclusions

In this paper, a system was developed and tested for the detection and classification of volcanoes from satellite images of the surface of the planet Venus using the YOLOv5 algorithm. For successful model training, it is necessary to have a large number of images and labeled objects in the data set. However, with relatively small data sets such as the Magellan dataset, it is possible to artificially increase the number of images to properly train the model. This is achieved by the joint use of classic data augmentation techniques (e.g., image rotation or mirroring) and more advanced methods (such as mosaic augmentation) in the manner presented in this paper. When performing data augmentation, it is important to take into account which methods are used, because otherwise unwanted features can be introduced into the training data set that result in worse model results. By testing the system on test images, the model gave satisfactory results considering the limitations provided by the dataset itself. Further improvements may be achieved by detailed tuning of hyperparameters. However, that was not the goal of the current article. Furthermore, the next step of research could be to develop an algorithm that divides the image into the N×N dimension grid cell in order to artificially increase image resolution. The goal of the proposed method is to try to increase the precision of the model in the detection and classification of objects of very small dimensions (around 20 × 20 pixels). In conclusion, this developed system performed successful and could find its application in mapping the surfaces of planets for the purposes of scientific research.

## Figures and Tables

**Figure 1 sensors-23-01224-f001:**
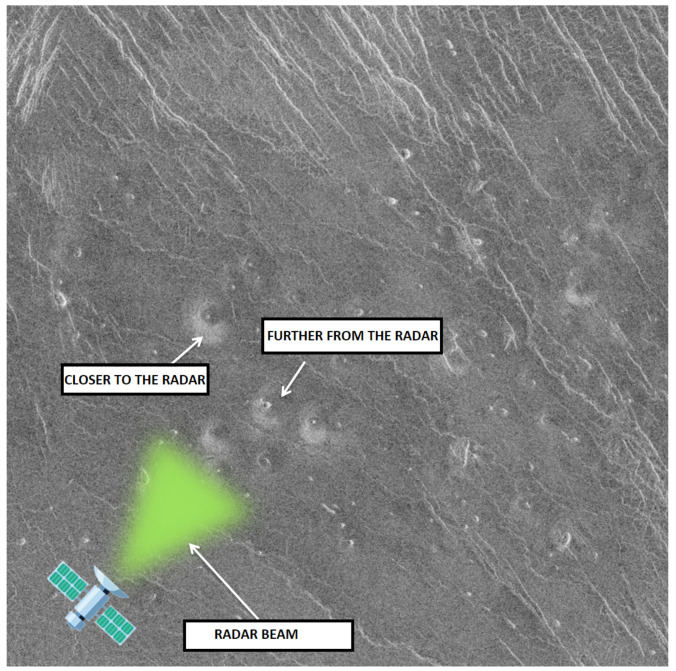
The example of a radar signature [8].

**Figure 2 sensors-23-01224-f002:**
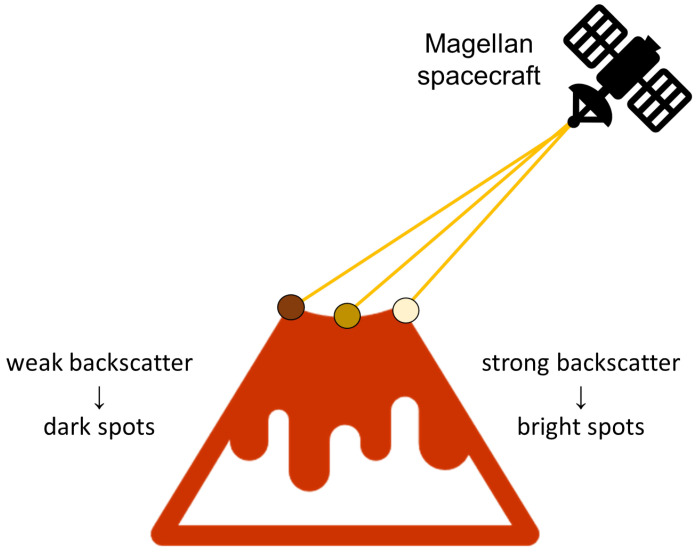
The reflection of the radar beam from the volcano [8].

**Figure 3 sensors-23-01224-f003:**
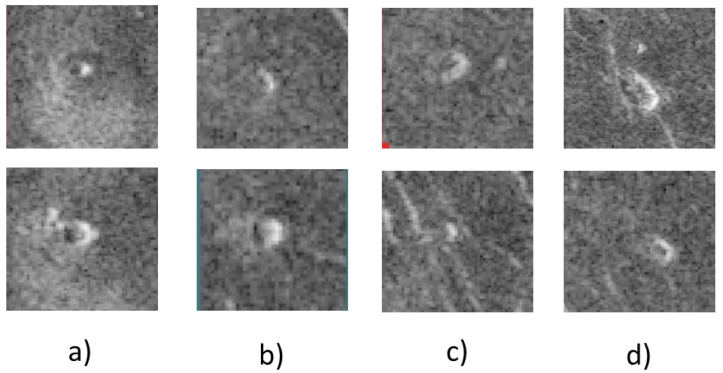
Defined volcano classes in the dataset: (**a**) Category 1, (**b**) Category 2, (**c**) Category 3, (**d**) Category 4 [8].

**Figure 4 sensors-23-01224-f004:**
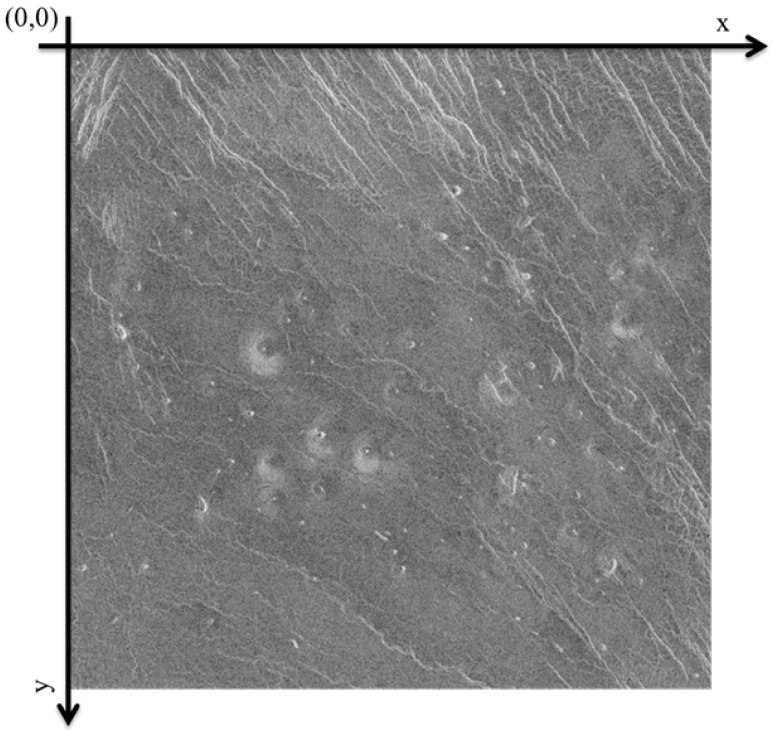
Image coordinate system.

**Figure 5 sensors-23-01224-f005:**
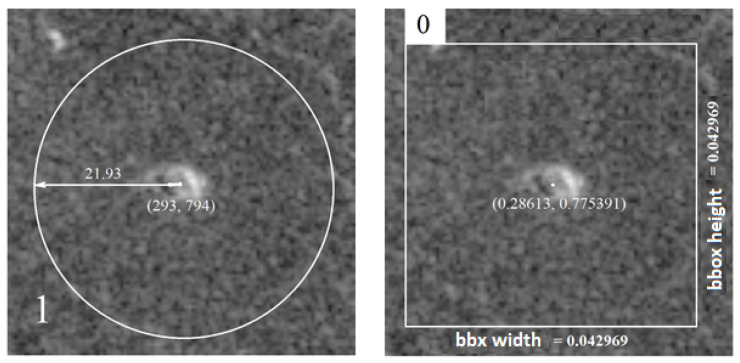
Difference between Magellan (**left**) and YOLO (**right**) annotations.

**Figure 6 sensors-23-01224-f006:**
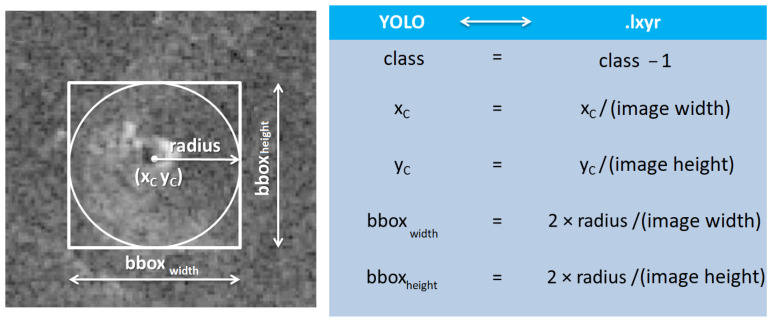
The method of transferring labels from Magellan format to YOLO format.

**Figure 7 sensors-23-01224-f007:**
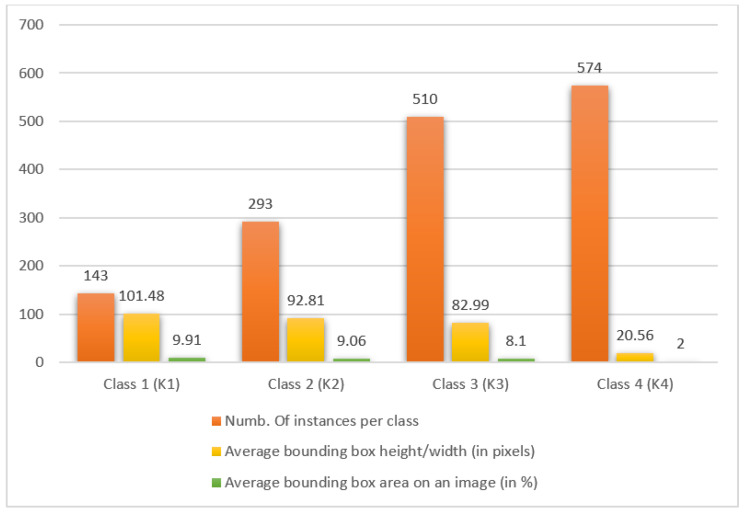
Analysis graph of the Magellan data set.

**Figure 8 sensors-23-01224-f008:**
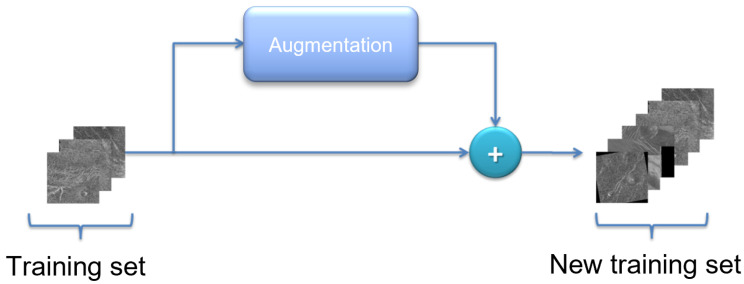
Schematic representation of dataset augmentation.

**Figure 9 sensors-23-01224-f009:**
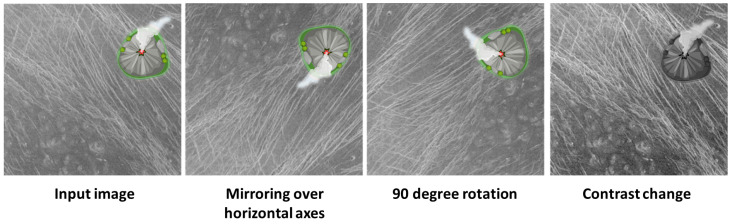
Examples of standard dataset augmentation techniques.

**Figure 10 sensors-23-01224-f010:**
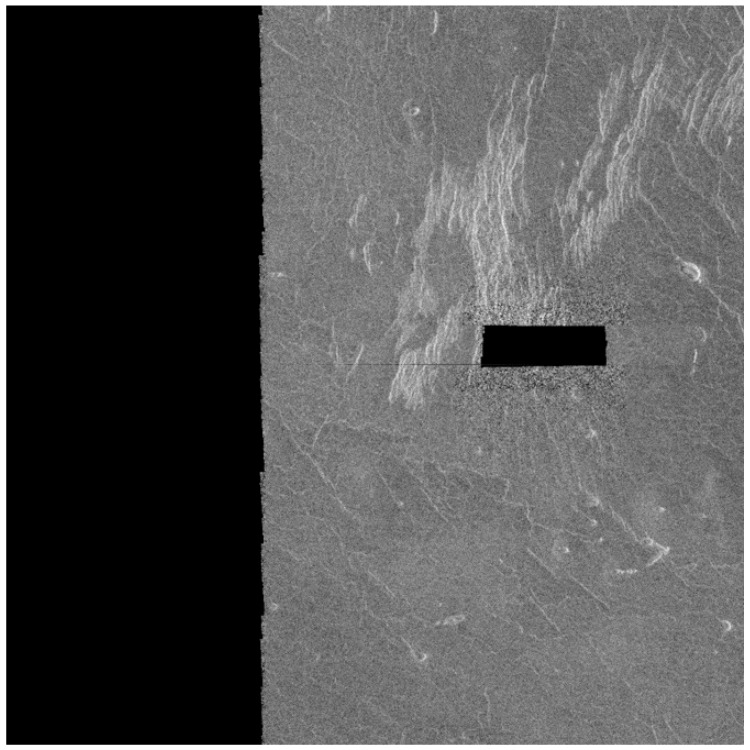
Examples of an image with a “black zone” [8].

**Figure 11 sensors-23-01224-f011:**
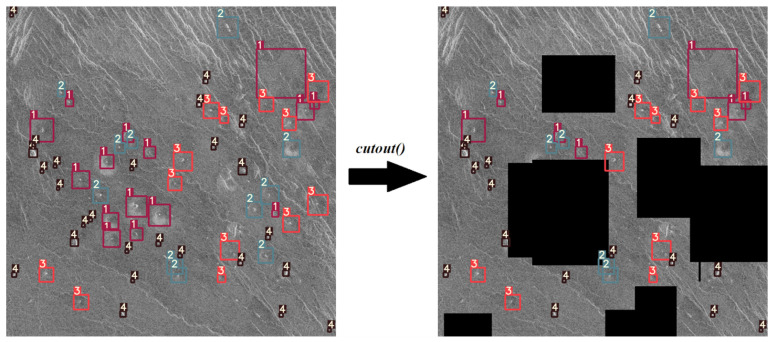
Example of an cutout agumentation.

**Figure 12 sensors-23-01224-f012:**
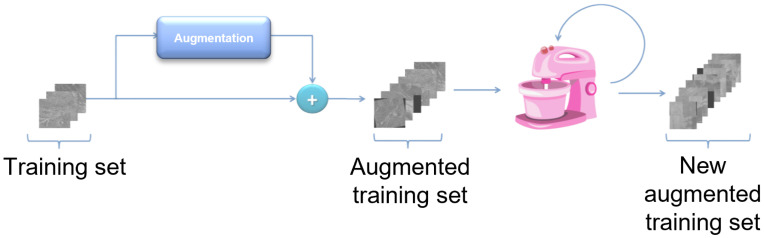
Schematic representation of augmentation pipeline.

**Figure 13 sensors-23-01224-f013:**
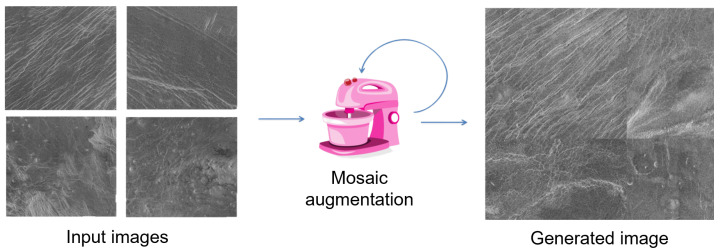
Concept of mosaic augmentation technique.

**Figure 14 sensors-23-01224-f014:**
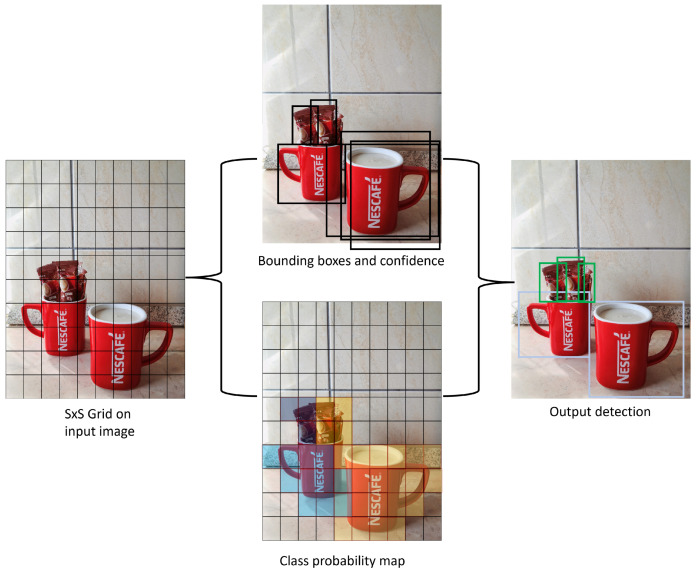
An ilustration on YOLOv5-based object detection.

**Figure 15 sensors-23-01224-f015:**
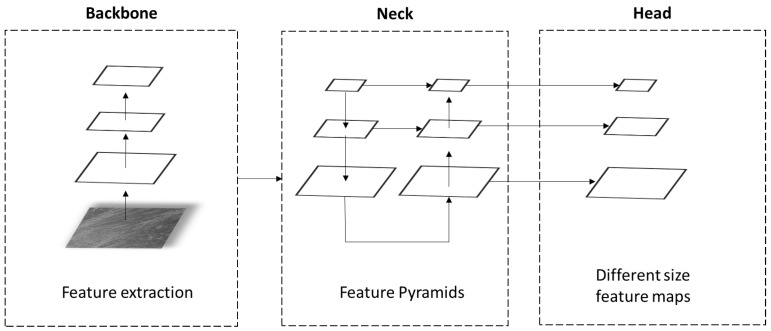
YOLOv5 structure.

**Figure 16 sensors-23-01224-f016:**
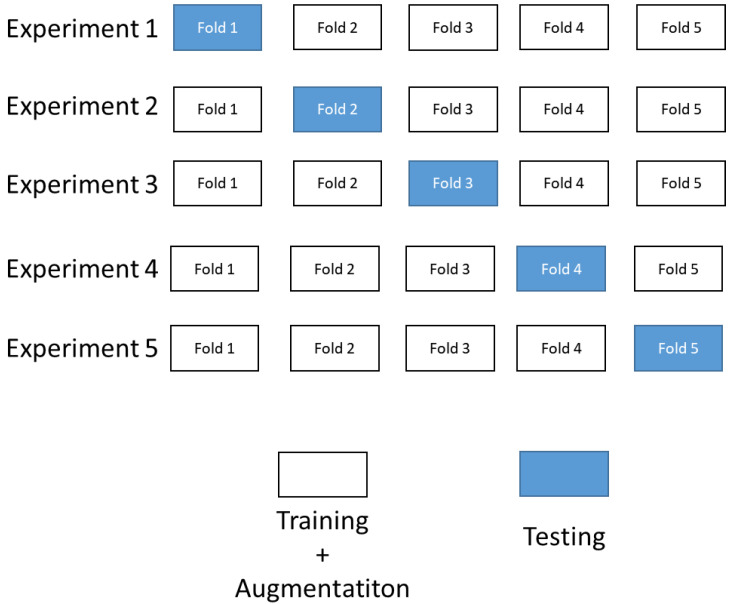
Illustration of a cross-validation procedure.

**Figure 17 sensors-23-01224-f017:**
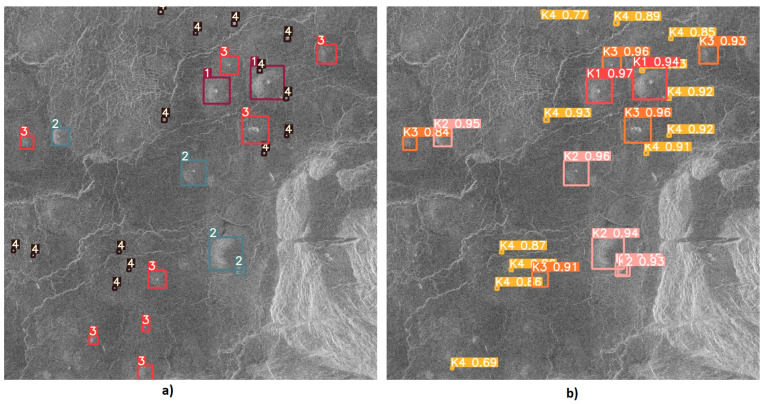
Input and output image used for testing the model-1: (**a**) Input image with given objects for detection. (**b**) Output image with detected objects.

**Figure 18 sensors-23-01224-f018:**
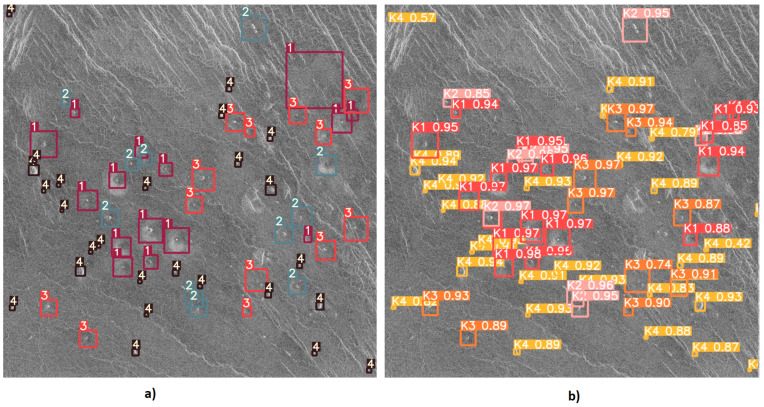
Input and output image used for testing the model-2: (**a**) Input image with given objects for detection. (**b**) Output image with the detected objects.

**Figure 19 sensors-23-01224-f019:**
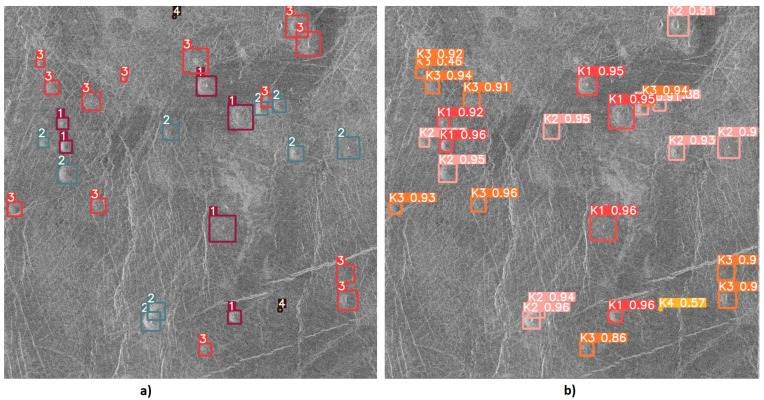
Input and output image used for testing model-3: (**a**) Input image with given objects for detection. (**b**) Output image with the detected object.

**Table 1 sensors-23-01224-t001:** Amount of data from existing and upcoming telescopes [6].

Telescope (year)	Data Rate (bytes/night)
VLT (1998)	10 GB
SDSS (2000)	200 GB
VISTA (2009)	315 GB
LSST (2019)	30 TB
TMT (2022)	90 TB

**Table 2 sensors-23-01224-t002:** Example of LXYR anotation file format.

Class	*X* Center	*Y* Center	Radius
1	273	720	21.3
2	130	450	50.2
1	423	123	70.1

**Table 3 sensors-23-01224-t003:** Example of YOLO annotation file format.

Class	*X* Center	*Y* Center	Width	Height
0	0.286	0.775	0.429	0.429
1	0.127	0.439	0.098	0.098
0	0.413	0.120	0.137	0.137

**Table 4 sensors-23-01224-t004:** YOLOv5 hyperparameters.

Hyperparameter	Value
Number of iterations	100
Optimizer	SGD
Input image resolution	1536 × 1536 pixels
Batch size	8
lr0	0.01
lr1	0.01
momentum	0.937
weight_decay	0.0005
warmup_epochs	3.0
warmup_momentum	0.8
warmup_bias_lr	0.1
box	0.05
cls	0.5
cls${}_{\mathrm{pw}}$	1.0
obj	1.0
obj${}_{\mathrm{pw}}$	1.0
iou${}_{t}$	0.2
anchor	4.0
fl${}_{\mathrm{gamma}}$	0.0
hsv${}_{h}$	0.015
hsv${}_{s}$	0.7
hsv${}_{v}$	0.4
degrees	0.0
translate	0.1
scale	0.5
shear	0.0
perspective	0.0
flipud	0.0
fliplr	0.5
mosaic	1.0
mixup	0.0
copy_paste	0.0

**Table 5 sensors-23-01224-t005:** Cross-validation results of a base model.

Metric	Experiment 1	Experiment 2	Experiment 3	Experiment 4	Experiment 5
K1 Precision	0.496	0.455	0.581	0.455	0.272
K1 Recall	0.493	0.483	0.600	0.483	0.500
K1 mAP@0.5	0.421	0.425	0.598	0.488	0.317
K2 Precision	0.252	0.294	0.249	0.294	0.232
K2 Recall	0.265	0.421	0.500	0.421	0.241
K2 mAP@0.5	0.151	0.241	0.182	0.204	0.162
K3 Precision	0.19	0.289	0.230	0.289	0.309
K3 Recall	0.267	0.430	0.490	0.430	0.324
K3 mAP@0.5	0.107	0.241	0.281	0.194	0.304
K4 Precision	0.231	0.348	0.364	0.348	0.364
K4 Recall	0.505	0.456	0.444	0.456	0.446
K4 mAP@0.5	0.251	0.284	0.290	0.248	0.345
Precision (all)	0.292	0.346	0.356	0.346	0.294
Recall (all)	0.382	0.448	0.509	0.448	0.378
mAP@0.5 (all)	0.233	0.298	0.338	0.284	0.282
F1-Score (all)	0.320	0.390	0.410	0.360	0.330

**Table 6 sensors-23-01224-t006:** Base model evaluation scores.

Metric	Average Value	Standard Deviaton
K1 Precision	0.457	0.101
K1 Recall	0.524	0.043
K1 mAP@0.5	0.449	0.092
K2 Precision	0.260	0.021
K2 Recall	0.356	0.09
K2 mAP@0.5	0.188	0.032
K3 Precision	0.256	0.042
K3 Recall	0.368	0.080
K3 mAP@0.5	0.225	0.070
K4 Precision	0.316	0.054
K4 Recall	0.449	0.036
K4 mAP@0.5	0.284	0.035
Precision (all)	0.322	0.026
Recall (all)	0.424	0.042
mAP@0.5 (all)	0.284	0.033
F1-Score (all)	0.362	0.034

**Table 7 sensors-23-01224-t007:** Cross-validation results of a base model + classical augmentation techniques.

Metric	Experiment 1	Experiment 2	Experiment 3	Experiment 4	Experiment 5
K1 Precision	0.530	0.503	0.644	0.467	0.546
K1 Recall	0.500	0.579	0.686	0.484	0.500
K1 mAP@0.5	0.503	0.473	0.643	0.450	0.531
K2 Precision	0.271	0.228	0.178	0.338	0.237
K2 Recall	0.309	0.382	0.250	0.446	0.276
K2 mAP@0.5	0.180	0.190	0.172	0.239	0.194
K3 Precision	0.228	0.412	0.305	0.257	0.352
K3 Recall	0.313	0.355	0.308	0.143	0.338
K3 mAP@0.5	0.144	0.328	0.246	0.113	0.256
K4 Precision	0.310	0.471	0.364	0.237	0.444
K4 Recall	0.434	0.353	0.318	0.400	0.373
K4 mAP@0.5	0.303	0.319	0.277	0.265	0.360
Precision (all)	0.335	0.403	0.373	0.325	0.444
Recall (all)	0.389	0.417	0.405	0.368	0.373
mAP@0.5 (all)	0.283	0.327	0.335	0.267	0.335
F1-Score (all)	0.360	0.410	0.390	0.340	0.410

**Table 8 sensors-23-01224-t008:** Base model + classical augmentation techniques evaluation scores.

Metric	Average Value	Standard Deviaton
K1 Precision	0.538	0.059
K1 Recall	0.549	0.076
K1 mAP@0.5	0.520	0.067
K2 Precision	0.250	0.055
K2 Recall	0.333	0.072
K2 mAP@0.5	0.195	0.023
K3 Precision	0.311	0.065
K3 Recall	0.306	0.084
K3 mAP@0.5	0.217	0.078
K4 Precision	0.375	0.072
K4 Recall	0.376	0.039
K4 mAP@0.5	0.305	0.033
Precision (all)	0.376	0.044
Recall (all)	0.390	0.018
mAP@0.5 (all)	0.309	0.028
F1-Score (all)	0.362	0.034

**Table 9 sensors-23-01224-t009:** Cross-validation results of a base model + augmentation pipeline.

Metric	Experiment 1	Experiment 2	Experiment 3	Experiment 4	Experiment 5
K1 Precision	0.916	0.863	0.880	0.948	0.770
K1 Recall	0.857	0.895	0.886	0.871	0.900
K1 mAP@0.5	0.906	0.932	0.924	0.927	0.846
K2 Precision	0.961	0.933	0.892	0.870	0.874
K2 Recall	0.857	0.750	0.731	0.823	0.717
K2 mAP@0.5	0.824	0.835	0.823	0.878	0.815
K3 Precision	0.872	0.935	0.931	0.914	0.895
K3 Recall	0.822	0.855	0.828	0.688	0.649
K3 mAP@0.5	0.873	0.900	0.886	0.792	0.762
K4 Precision	0.892	0.928	0.885	0.907	0.847
K4 Recall	0.778	0.757	0.590	0.728	0.554
K4 mAP@0.5	0.796	0.819	0.702	0.775	0.686
Precision (all)	0.910	0.915	0.897	0.910	0.847
Recall (all)	0.793	0.814	0.759	0.777	0.705
mAP@0.5 (all)	0.850	0.872	0.834	0.843	0.777
F1-Score (all)	0.850	0.86	0.820	0.840	0.760

**Table 10 sensors-23-01224-t010:** Base model + augmentation pipeline evaluation scores.

Metric	Average Value	Standard Deviaton
K1 Precision	0.974	0.06
K1 Recall	0.882	0.015
K1 mAP@0.5	0.907	0.032
K2 Precision	0.906	0.035
K2 Recall	0.747	0.039
K2 mAP@0.5	0.835	0.022
K3 Precision	0.909	0.023
K3 Recall	0.768	0.083
K3 mAP@0.5	0.842	0.055
K4 Precision	0.891	0.026
K4 Recall	0.681	0.009
K4 mAP@0.5	0.756	0.052
Precision (all)	0.896	0.025
Recall (all)	0.769	0.037
mAP@0.5 (all)	0.835	0.032
F1-Score (all)	0.826	0.036

## Data Availability

“Magellan dataset” at http://archive.ics.uci.edu/ml/datasets/volcanoes+on+venus+-+jartool+experiment (accessed on 15 January 2022).

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
