# Peer review of "Localization and Classification of Venusian Volcanoes Using Image Detection Algorithms"

_sensors, 2023, doi:10.3390/s23031224_

Round 1
Reviewer 1 Report
The article is clearly written and divided into well-defined chapters. I found several formal errors in the text that I recommend to correct:
Figure 2 - here I am missing the link to the source from which it was taken and the image could be a little bigger or brighter for better recognition.
Figure 11 - Here again, the source from which it was taken is missing.
319 - "valuesof IoU..." here split the text into "values of"
321 - "valuesof IoU..." here split the text into "values of"
336 - "YOLOV5m6" ...... modify here to lowercase "v" "YOLOv5m6"
Table 5. - Reduce to page width.
348 - Delete the word "Table" once.
379 - 380 - delete "it can" once
Figure 15. - at the end of the description, the plural of the word object - "objects".
Figure 16. - at the end of the description, the plural of the word object - "objects".
For all the darker images, I would recommend lightening for easier recognition.
For the meaning of Augmentation, I am not quite sure if the term Enlargement should not be used. Here I will leave it to the careful consideration of the authors.
Reviewer 2 Report
In this paper, YOLO algorithm is applied to the location and classification of Venus volcano in Magellan dataset, which contains 134 satellite images of Venus surface, and each volcano is marked with a circular label. The original labels are first converted to the appropriate YOLO format. Then, aiming at the problem that the number of images in the dataset is relatively small, the deterministic enhancement technology is adopted. The experimental results show that the method is effective. However, the following problems need to be improved:
1) The YOLO network model and parameters shall be introduced as necessary.
2) There are some grammar and expression errors.
Reviewer 3 Report
- Using the hold out method with 70/30 split is insufficient with such small dataset. Extensive cross-validation need to be performed.
- Are the results for the validation set or testing test?
- What were the backbone CNNs and detections heads locations?
- Augementation with this small dataset will lead to data leaking (i.e., overfitting and inflatted results). Moreover, the type of augmentation effects do not make sense, e.g., a volcano upsidedown.
- The precision-recall curves need to be provided.
- Line 47, typo, reference number should be before the period.
- Line 59, no need to capitalize the word "Principal".
- Lines 131-134 capitalize the first letter of the first word of the sentence.
- Similar studies utilizing Yolo can be cited so that to established the trustworthiness of the models and can provide reliabilit to baseline settings, see Detection of K-complexes in EEG signals using deep transfer learning and YOLOv3. Cluster Comput (2022). https://doi.org/10.1007/s10586-022-03802-0.
- The table of abbreviations as required by the journal template is missing.
Round 2
Reviewer 3 Report
The authors addressed my comments.